# Increased Influenza Vaccination Coverage among Members of the Athens Medical Association Amidst COVID-19 Pandemic

**DOI:** 10.3390/vaccines10050797

**Published:** 2022-05-18

**Authors:** Georgios Marinos, Dimitrios Lamprinos, Panagiotis Georgakopoulos, Evangelos Oikonomou, Georgios Zoumpoulis, Gerasimos Siasos, Dimitrios Schizas, Christos Damaskos, Nikolaos Garmpis, Anna Garmpi, George Patoulis, Fotios Patsourakos, Ioannis Datseris, Efstathios Tsoukalos, Ioannis D. Anyfantis, Dimitrios Papagiannis, Emmanouil K. Symvoulakis, Georgios Rachiotis

**Affiliations:** 1Department of Hygiene, Epidemiology and Medical Statistics, School of Medicine, National and Kapodistrian University of Athens, 11527 Athens, Greece; 2Emergency Department, Laiko General Hospital, 11527 Athens, Greece; dimitrislamprinos@gmail.com (D.L.); panos.k.georgakopoulos@gmail.com (P.G.); gzoumpoulis@yahoo.gr (G.Z.); 3Third Department of Cardiology, Thoracic Diseases General Hospital Sotiria, School of Medicine, National and Kapodistrian University of Athens, 11527 Athens, Greece; boikono@gmail.com (E.O.); gsiasos@med.uoa.gr (G.S.); 4First Department of Cardiology, Hippokration General Hospital, School of Medicine, National and Kapodistrian University of Athens, 11527 Athens, Greece; 5First Department of Surgery, Laikon General Hospital, School of Medicine, National and Kapodistrian University of Athens, 11527 Athens, Greece; dschizas@med.uoa.gr; 6N.S. Christeas Laboratory of Experimental Surgery and Surgical Research, School of Medicine, National and Kapodistrian University of Athens, 11527 Athens, Greece; x_damaskos@yahoo.gr; 7Second Department of Propaedeutic Surgery, Laiko General Hospital, Medical School, National and Kapodistrian University of Athens, 11527 Athens, Greece; nikosg22@hotmail.com; 8First Department of Propaedeutic Internal Medicine, Laiko General Hospital, Medical School, National and Kapodistrian University of Athens, 11527 Athens, Greece; annagar@windowslive.com; 9Athens’s Medical Association, 11527 Athens, Greece; gipattt@gmail.com (G.P.); fnpatsourakos@gmail.com (F.P.); jodats13@me.com (I.D.); stathisnapoleon.tsoukalos@gmail.com (E.T.); 10European Agency for Safety and Health at Work (EU-OSHA), Prevention and Research Unit, 48003 Bilbao, Spain; anyfantis@osha.europa.eu; 11Public Health & Vaccines Laboratory, Faculty of Nursing, School of Health Science, University of Thessaly, 41110 Larissa, Greece; dpapajon@gmail.com; 12Clinic of Social and Family Medicine, Faculty of Medicine, University of Crete, 71500 Heraklion, Greece; symvouman@yahoo.com; 13Department of Hygiene and Epidemiology, Faculty of Medicine, University of Thessaly, 41222 Lariss, Greece; grach@uth.gr

**Keywords:** influenza, vaccination, physicians, health care workers, COVID-19

## Abstract

Healthcare workers are at high risk of influenza virus infection as well as of transmitting the infection to vulnerable patients who may be at high risk of severe illness. The aim of this cross-sectional study was to investigate the prevalence and related factors of influenza vaccination coverage (2020–2021flu season), among members of the Athens Medical Association in Greece. This survey employed secondary analysis data from a questionnaire-based dataset on COVID-19 vaccination coverage and associated factors from surveyed doctors, registered within the largest medical association in Greece. All members were invited to participate in the anonymous online questionnaire-based survey over the period of 25 February to 13 March 2021. Finally, 1993 physicians (60% males; 40% females) participated in the study. Influenza vaccination coverage was estimated at 76%. Logistic regression analysis demonstrated that older age (OR = 1.02; 95% C.I. = 1.01–1.03), history of COVID-19 vaccination (OR = 2.71; 95% C.I. = 2.07–3.56) and perception that vaccines in general are safe (OR = 16.49; 95% C.I. = 4.51–60.25) were found to be independently associated factors with the likelihood of influenza vaccination coverage. Public health authorities should maximize efforts and undertake additional actions in order to increase the percentage of physicians/health care workers (HCWs) being immunized against influenza. The current COVID-19 pandemic offers an opportunity to focus on tailored initiatives and interventions aiming to improve the influenza vaccination coverage of HCWs in a spirit of synergy and cooperation.

## 1. Introduction

There is sufficient evidence that influenza epidemics can be the cause of considerable morbidity, hospitalization and death, especially during the winter months. While influenza infection is usually mild and uncomplicated, it can cause severe disease, particularly among the elderly, persons living in long-term care facilities and other vulnerable groups, such as pregnant women, persons with underlying medical conditions and young children. Influenza symptoms range from fever, cough, body aches and headache, to severe primary viral pneumonia, which can be complicated by bacterial superinfection and exacerbation of underlying chronic conditions. Vaccination against influenza is safe and the most effective means of preventing infection and severe outcomes caused by influenza viruses, including morbidity, severe disease, hospitalization and death. As noted above, influenza usually represents a mild self-limiting disease of the respiratory system. However, there is a risk of severe complications for respiratory, cardiovascular and neurological systems [1,2]. Health care workers (HCWs), including care workers, in hospitals’ long-term care facilities were recommended as one of the highest priority groups for influenza vaccination during the COVID-19 pandemic, and they were recommended to receive annual influenza vaccination due to having a higher risk of infection and potential role in the transmission of influenza to vulnerable patient groups [3,4,5].In addition, the WHO recommends the immunization of HCWs against influenza in order to control and minimize work absenteeism due to influenza and disruption to the workforce and keep services safe and personnel protected [6,7].There are reports indicating a large variation in influenza vaccination coverage among HCWs, ranging from 2.6% to 99.5%, with a median of 29.5% (2014/2015 data), with very few countries reporting a high vaccination coverage (>75%) [8].Most countries reported suboptimal influenza vaccination coverage rates (<40%). In the context of the COVID-19 pandemic, there were concerns around the world that severe acute respiratory syndrome coronavirus 2 (SARS-CoV-2) and influenza might cocirculate and drive a dual infective synergism, which would further increase morbidity, use of healthcare services and mortality [9]. In fact, the prevalence of influenza at a global scale was reported as considerably low over the 2020–2021 flu season [10]. This fact could be attributed to the wide implementation of primary prevention measures (e.g., use of face masks, physical distancing, restrictions), and possibly to the increased flu vaccination coverage [10,11]. On the bases of the above, it is worth investigating flu vaccination coverage among HCWs amidst the COVID 19 pandemic. The aim of this study was to investigate the prevalence and related factors of influenza vaccination coverage (2020–2021 flu season) among the members of the Athens Medical Association (Iatrikos Syllogos Athinon; I.S.A.) in Greece.

## 2. Materials and Methods

This cross-sectional study (survey) employed secondary analysis data from a questionnaire-based dataset on COVID-19 vaccination coverage and associated factors, which were reported from members of the Athens Medical Association in Greece (I.S.A.) [12]. In particular, a survey was conducted over the period 25 February to 13 March 2021. All members of I.S.A. were invited to participate in an anonymous online questionnaire-based survey [10]. The questionnaire contained items on demographic background of the participants (gender, age, and occupational characteristics); perceptions of the importance of vaccinations (Vaccines are important for public health); and attitudes towards, safety (In general, vaccines are safe), and effectiveness (In general, vaccines are effective) of the vaccines. In addition, the questionnaire included the following question aiming to record the influenza vaccination coverage for the 2020–2021flu season: “Have you been vaccinated with the influenza vaccine (season 2020–2021)?” (Yes/No; 1 = yes, 0 = no).

COVID-19 vaccination coverage was evaluated by using the following item: “Are you vaccinated against COVID-19?” Participants were asked to rate on a four-point Likert scale the importance, safety, and effectiveness of vaccinations (fully agree/agree/disagree/fully disagree). For the purposes of the multivariable analysis (logistic regression analysis), these four response categories were further restricted to two (fully agree/agree vs. disagree/fully disagree).

All participants gave consent for participation by accepting to complete an anonymous e-questionnaire, after receiving introductory information on the study’s aims. Responses were anonymously collected and no content related to the respondents’ identity was retrievable. The protocol of the study was approved by the Board of the Athens Medical Association (Code: 18.02.21).

### Statistical Analysis

Relative (%) and absolute frequencies were presented for categorical variables, while continuous variables were presented as mean ± standard deviation. Chi-squared test (χ^2^) was used for the univariate analysis of categorical variables. *t*-test was used for the univariate analysis of continuous variables. Variables with a *p* value < 0.25 in the univariable analysis were included in a stepwise binary logistic regression analysis model (logistic regression analysis was performed using the backward conditional method), in order to explore potential independent associations with influenza vaccination coverage. Odds ratios (ORs) and 95% confidence intervals (95%, C.I.) were calculated. Further more, the Hosmer-Lemeshow test was employed to evaluate the goodness-of-fit of the logistic regression model used. A *p* value < 0.05 was considered to indicate statistical significance. The level of statistical significance level was set at *p* value < 0.05. All data were analyzed with SPSS software (IBM SPSS Statistics for Windows, Version 26.0. Armonk, NY, USA: IBM Corporation).

## 3. Results

The total number of participants was 1993. The total number of members of the Athens Medical Association was 25,900 (response rate = 8%). Among the participants, 1192 (59.8%) were males and 801 (40.2%) females. The mean age was 52.9 years (SD = 10.73). Influenza vaccination coverage among participants was 76.4%. The results of the univariable analysis (Table 1) showed that gender and employment status were not significantly associated with influenza vaccination coverage. Older age and positive attitudes towards vaccinations (effectiveness, safety, value) were significantly associated with increased prevalence of influenza vaccination. Physicians who reported that they were vaccinated against COVID-19 demonstrated higher influenza vaccination coverage than their colleagues not vaccinated against COVID-19 (80% vs. 55%, *p* value < 0.001).

### Multivariable Analysis

Logistic regression analysis (results reported in Table 2) showed that older age (OR = 1.02; 95% C.I. = 1.01–1.03), history of COVID-19 vaccination (OR = 2.71; 95% C.I. = 2.07–3.56) and perception that vaccines in general are safe (OR = 16.49; 95% C.I. = 4.51–60.25) were independently associated with the likelihood of influenza vaccination coverage.

## 4. Discussion

In this study, we recorded influenza vaccination coverage and factors correlated with it, among members of the largest medical association in Greece in the era of the COVID-19 pandemic. We found a satisfactory rate of influenza vaccination coverage (76.4%). This figure confirmed the results of a previous study in Central Greece, which reported an influenza vaccination coverage at 74% among members of the local medical association and represented a considerable deviation from historical trends of influenza vaccination among Greek HCWs [13,14,15]. In particular, Rachiotis et al. reported that acceptance of the 2009 pandemic influenza A (H1N1) vaccine was estimated at 17% among HCWs from Central Greece [14]. In addition, Maltezou et al. in a nationwide study (2015–2018) reported a profound increase in influenza vaccination coverage among HCWs. In fact, the vaccination coverage increased from 10.9% to 24.9% among HCWs in hospitals, and from 24.3% to 40.2% among primary care HCWs [15]. Further, there are several lines of evidence from European countries which indicate a possible positive impact of the COVID-19 pandemic on improvements in influenza vaccination coverage among HCWs. In particular, Scardina and co-authors reported a notable increase in influenza vaccination coverage from 11.9% (2019–2020flu season) to 39.5% (2020–2021flu season) among HCWs of a university hospital in Italy [16]. Štěpánek et al. in a cross-sectional study among HCWs in Czech Republic observed an elevated motivation for influenza vaccination during the 2020–2021flu season, as they concluded that COVID-19 contributed to an increased influenza vaccination uptake among HCWs and the fear of contracting it together with influenza was the most frequent driver of vaccination [17].

Similarly, Stockeler et al. reported that influenza vaccination coverage was elevated from 31% in the 2016/2017 flu season to 59% in 2020/2021 among HCWS in Bavaria, south-eastern Germany [18]. In addition, a study among HCWs in Ireland demonstrated that more HCWs were interested in influenza vaccination (2020–2021flu season) than ever before [19]. The COVID-19 pandemic has had some influence on staff’s likelihood of being vaccinated [19].

Interestingly, history of COVID-19 vaccination was an independent factor associated with influenza vaccination. This finding is aligned with the results of a general population study conducted in the United Kingdom, which indicated that the COVID-19 pandemic stimulated an increased uptake of influenza vaccination in 2020–2021 [20]. In our study, uptake of COVID-19 vaccination would be considered as an indirect indicator of perceived risk of COVID-19 infection and related complications. Further, there is evidence that dual infection with COVID-19 and influenza elevates the risk of death [21], and possibly, this eventuality could lead to an elevated uptake of influenza vaccination among both the general population and HCWs.

As mentioned above, the activity of influenza in Europe was low during the period 2020–2021. In particular, as of week 8/2021, a small number (*n* = 712) of influenza cases across the European region have been registered [10]. Nevertheless, there is some evidence (week 4, 2022) that indicates local, regional and widespread influenza occurrence in some European countries [22]. However, because it is largely unclear how influenza will evolve, it is of importance to increase vaccination coverage [23] and to better learn how to motivate individuals and population groups to be vaccinated. The emerging variable, from the multivariate analysis, in regard to vaccine safety in general appears to significantly strengthen the likelihood of motivating flu vaccination uptake. However, it is not sufficient to justify previous poor influenza vaccine coverage. Most of the participating physicians probably had a similar perception on safety a few years ago. Psychological inertia is the propensity to maintain a known status or a default option unless a psychological motivation to drive intervention or its rejection emerges [24]. It appears that COVID-19 vaccine acceptance may offer a motivation to reject a prior status of inertia towards influenza vaccines or others. This observation should be proved with further health behavioral research for future knowledge and planning. Lastly, increasing age was associated with increased likelihood of reported influenza vaccination coverage among physicians of Athens Medical Association. This finding is in line with the results of previous studies [14] and may be related to the increased prevalence of physicians with comorbidities among those of older age. Vaccine hesitancy consists of three pillars: complacency (perception of low risk), confidence and convenience. We can speculate that increasing age among physicians may enhance risk-perception-related complacency. There are several lines of evidence which indicate that influenza vaccination coverage among the elderly has increased in Europe [25]. On the other hand, there are studies that fail to report age as an independent predictor of influenza vaccination among HCWs [26].

### Limitations and Strengths

The present study has several shortcomings, which should be taken into consideration prior to the interpretation of the results. First, our study is of a descriptive, cross-sectional nature and it is not possible for authors to provide the reader with causal associations between investigated risk factors and flu vaccination coverage. Second, due to the questionnaire-based structure of our study, it is likely that recall or information bias may have occurred. Further, healthcare professionals who were not interested in the vaccination or were against vaccination may have decided not to participate in our survey and, thus, it is possible that selection bias occurred. In addition, we should note the low response rate of our descriptive study. Notwithstanding, it should be underlined that online surveys’ low response rates have been noted as a common concern for many researchers. We were also not able to obtain data from non-respondents and this inhibits our ability to compare and offer more generalizable explanations. Moreover, our sample included only physicians and we are not able to report data for other groups of HCWs, such as nurses. Lastly, it should be mentioned that our study was based on secondary data and the primary outcome of interest was COVID-19 vaccination. Consequently, we did not directly assess the attitudes of the participants towards influenza-related vaccines. Notwithstanding this limitation, influenza vaccines have a long history of application and could be considered as “old” vaccines. Consequently, we may consider that our questionnaire items aiming to assess the attitudes of physicians towards vaccines, in general, may have indirectly included influenza vaccines too [13].

## 5. Conclusions

In conclusion, here, we report an increased rate of influenza vaccination coverage among Greek physicians working in the capital city of the country. History of COVID-19 vaccination was independently associated with the likelihood of influenza vaccination coverage. The current COVID-19 pandemic indicates that the attitudes and practices of physicians towards influenza vaccination could be positively modified and this represents an opportunity to focus on snowball messages and informative campaign initiatives to further improve the influenza vaccination coverage of HCWs as a whole [27]. Public and occupational health authorities should undertake additional actions, taking advantage of COVID-19’s momentum in order to further improve influenza vaccination rates among physicians and health care providers, using research findings to guide their decisions. In this context, several types of tailored initiatives including educational, increased awareness projects, and “soft” interventions may contribute to improvements in influenza vaccination coverage in a spirit of synergy and cooperation [28].

## Figures and Tables

**Table 1 vaccines-10-00797-t001:** Univariable analysis of influenza vaccination coverage (season: 2020–2021).

Risk Factor	Influenza Vaccination Coverage		
	Yes (%)	No (%)	*p* Value
Gender			
Male	921 (77.3)	271 (22.7)	0.25
Female	601 (75)	200 (25)	
Age (Years, Mean, SD)	53.2 ± (10.7)	51.1 (10.9)	0.001
Employment status			
Working in the public sector (NHS, Army, Universities)	397 (77.5)	115 (22.5)	0.469
Working in the private sector	1125 (76)	356 (24)	
Vaccines are important for Public Health			
Fully Agree/Agree	1519 (76.5)	467 (23.5)	0.037
Fully disagree/Disagree	3 (42.9)	4 (57.1)	
In general, vaccines are safe.			
Fully Agree/Agree	1517 (77.4)	443 (22.6)	<0.001
Fully disagree/Disagree	5 (15.2)	28 (84.8)	
In general, vaccines are effective.			
Fully Agree/Agree	1516 (76.8)	458 (23.2)	<0.001
Fully disagree/Disagree	6 (31.6)	13 (68.4)	
Have you been vaccinated against COVID-19?			
Yes	1361 (80)	340 (20)	<0.001
No	161 (55.1)	131 (44.9)	

**Table 2 vaccines-10-00797-t002:** Multivariable analysis of influenza vaccination coverage (2020–2021).

Independent Variable	AOR **	95% C.I. ***	*p* Value
Age *	1.02		
	1.00 (ref)	1.01–1.03	<0.001
COVID-19 vaccination			
Yes	2.71	2.07–3.56	<0.001
No	1.00 (ref)		
In general, vaccines are safe			
Yes	16.49	4.51–60.25	<0.001
No	1.00 (ref)		

* Continuous variable; ** AOR: Adjusted Odds Ratio; *** 95% Confidence Interval.

## Data Availability

Data would be available upon reasonable request.

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
