# Peer review of "Increased Influenza Vaccination Coverage among Members of the Athens Medical Association Amidst COVID-19 Pandemic"

_vaccines, 2022, doi:10.3390/vaccines10050797_

Round 1
Reviewer 1 Report
In this short manuscript, the authors aimed to investigate the prevalence and related factors of influenza vaccination coverage (season 2020-2021) among the members of the Athens Medical Association (Iatrikos Syllogos Athinon; I.S.A.) in Greece.
This is an important topic as there has been little influenza activity around the world since April 2020 and finding ways to achieve high vaccination coverage rates will be important in the future, when influenza will circulate again.
Some comments below:
- The introduction is a bit short and probably does not give a sufficient background on influenza vaccination to the reader. One idea to make it more exhaustive is adding a general paragraph on the importance of influenza vaccination and influenza vaccination coverage in different age groups/categories AND in these pandemic times. Please find below some important references you may want to use [1-2].
- Were all the participants medical doctors (as the association is named Athens Medical Association) or healthcare professionals in general? This should be clearly stated. If the participants were all doctors, it should be stated in the limitations that even considering the group of HCWs there is a potential selection bias, as medical doctors could represent a different, homogeneous group.
- Despite mentioning the paper you have already published that was based on this questionnaire, I suggest giving more details about it, as the method section is not so clear (i.e. what were the questions? What could the participants respond?).
- The discussion is interesting and nicely written. It is important that you mentioned the impact that the COVID-19 pandemic had on the uptake of influenza vaccination. Please find below another important reference that reinforces your message and you may want to cite [3] along with the UK observational study you have already mentioned.
- WHO Regional Office for Europe recommendations on influenza vaccination for the 2020/2021 season during the ongoing COVID-19 pandemic. https://apps.who.int/iris/bitstream/handle/10665/335721/WHO-EURO-2020-1141-40887-55342-eng.pdf?sequence=1&isAllowed=y
- Grech, V., & Borg, M. (2020). Influenza vaccination in the COVID-19 era. Early human development, 148, 105116.
- Del Riccio M, Lina B, Caini S, et al. Letter to the editor: Increase of influenza vaccination coverage rates during the COVID-19 pandemic and implications for the upcoming influenza season in northern hemisphere countries and Australia. Euro Surveill. 2021;26(50):2101143. doi:10.2807/1560-7917.ES.2021.26.50.2101143
Author Response
On behalf of the authors teams we would like to thank the reviewers for reviewing our paper entitled “Increased influenza vaccination coverage among members of the Athens Medical Association amidst COVID-19 pandemic”.

Reviewer 2 Report
I read the paper titled "Increased influenza vaccination coverage among members of the Athens Medical Association amidst COVID-19 pandemic", by Marinos G et al.
Some minor comments could improve the quality of manuscript.
Abstract and conclusions
the authors state that: Public health authorities should maximize efforts and undertake additional actions in order to increase the percentage of physicians/health care workers (HCWs) being immunized against influenza. This concept should be stressed more in the context of discussion.
Method and Results
The authors, should provide more details about the questionnaire. Indeed, it is not enough that they cite manuscript #10 (To insert table should be more appropriate). Does the questionnaire you prepared reveal patients' attitudes about getting vaccinated against infectious diseases?
DIs people who were vaccinated against influenza subsequently contract the disease? And what percentage of people (medical staff) who were not vaccinated against influenza, later contracted the respiratory disease?
Results: physicians who reported that they are vaccinated against COVID 19 demonstrated higher influenza vaccination coverage than their colleagues not 114 vaccinated against COVID-19 (80% vs 55%, p-value<0,001). Please, comment this sentence in the discussion section.
Author Response

(The authors gave the same response as above.)

Reviewer 3 Report
Review:
Increased influenza vaccination coverage among members of the Athens Medical Association amidst COVID-19 pandemic.
This study aimed to investigate the prevalence and related factors of influenza vaccination coverage (season 2020-2021), among the members of the Athens Medical Association, in Greece, using data from a questionnaire-based dataset on the COVID-19 vaccination coverage and associated factors from surveyed doctors, registered within the largest medical association.
The findings are important to health care workers in public health areas, particularly during the COVID-19 pandemic. The authors clearly stated the objectives, methods, contributions, and limitations of the study. The manuscript is straightforward to read. However the language could benefit from proofreading for grammar and clarity by a native English speaker or a professional proof-reader.
Introduction and Discussion section need to be enriched. Statistical analysis section needs to be improved. Some detailed comments for the authors to consider and perhaps incorporate into a revised version of the paper have been given below.
Specific comments:
Abstract:
Line 38: need to include the sample size, % of gender of the study.
Introduction:
Line 60-61: “Most countries reported suboptimal influenza vaccination coverage rates 60 (<40%)”: for which year or period or season? Is this conclusion from Ref [6]? If not, attach a ref.
The section provided influenza vaccination coverage among HCWs for 2014-2015 season. Suggest the authors provide some information on influenza vaccination coverage for just previous flu season in Europe, say 2018-2019, if data are available.
Materials and Methods
Line 76-77: provide some more information on the survey: type, sample size, response rate.
Line 82-83: suggest indicating specifically the main outcome was coded “1” for Yes, and “0” for No, which would be matched to the aOR in the Table 2.
Line 88: “For the purposes of the multivariate analysis”: suggest change “multivariate” to “multivariable” (whenever applied in the manuscript) as the two terminologies are different statistically.
Line 85: “Are you vaccinated against COVID-19?” (Yes/No): for which year?
Line 96: suggest the authors consult a biostatistician for advice on a revision of this section, and here are some points for reference:
- Suggest change “qualitative” to “categorical”, and “quantitative” to “continuous” in the manuscript.
- “Chi-square test” should be “Chi-squared test”.
- “Student’s t- test”: suggest change to “independent (or two) samples t-test”, given that all categorical factors were binary in this study. In addition, the authors need to indicate whether they assessed normality before performing the t-test.
- “Variables with a p value < 0.25”: need a ref for this variable-selection strategy.
- “a stepwise binary logistic regression analysis model”: suggest the authors indicate which specific SPSS stepwise strategy used for developing their final logistic model. In addition, whether the authors assessed the goodness-of-fit of the final model using Hosmer and Lemeshow test?
- “The level of statistical significance level was set at p = 0.05”: is not a correct sentence and suggest rewrite
- “Odds Ratio”: should be adjusted Odds Ratio (aOR), given that the authors included all variables (with a p value < 0.25) into their multiple regression model. Need to change this notation accordingly in the results section/table.
- Need to attach a citation for SPSS.
Results:
Line 110: “did not have a significant impact”: this study was only able to examine the association.
Line Table 1&2: suggest use the same decimal places. Need a footnote for aOR and C.I. Suggest use consistent term for explanatory/independent variables.
Line 112: “…were significantly associated with the prevalence of influenza vaccination: need to show the difference direction (e.g., higher prevalence rates, or increased coverage…).
Discussion
Line 128-130: whether the referred study was before the COVID 19 pandemic? In addition, suggest the authors list figures to show the change in trends of influenza vaccination after than before COVID 19 pandemic among Greek HCWs.
Line 156: how low? Suggest attach the prevalence of influenza in Europe.
No discussions for age were found.
Line 171: whether the low response rate of the survey could affect the representative of the study population or the generalizability of the study outcomes? Also the authors need to explain the reason of the wide 95%CI associated with vaccines perception (95% CI: 4.51-60.25).
Good luck!
Author Response

(The authors gave the same response as above.)

Round 2
Reviewer 1 Report
The manuscript has improved in quality and overall readability in the present form. Thank you for addressing the comments.
Good luck